# Semi-Supervised Group Emotion Recognition Based on Contrastive Learning

**Jiayi Zhang [1,†], Xingzhi Wang [1,†], Dong Zhang [1,\*] and Dah-Jye Lee [2]**

1   School of Electronics and Information Technology, Sun Yat-sen University, Guangzhou 510006, China
2   Department of Electrical and Computer Engineering, Brigham Young University, Provo, UT 84602, USA
\*   Correspondence: zhangd@mail.sysu.edu.cn
†   These authors contributed equally to this work.

**Abstract:** The performance of all learning-based group emotion recognition (GER) methods depends on the number of labeled samples. Although there are lots of group emotion images available on the Internet, labeling them manually is a labor-intensive and cost-expensive process. For this reason, datasets for GER are usually small in size, which limits the performance of GER. Considering labeling manually is challenging, using limited labeled images and a large number of unlabeled images in the network training is a potential way to improve the performance of GER. In this work, we propose a semi-supervised group emotion recognition framework based on contrastive learning to learn efficient features from both labeled and unlabeled images. In the proposed method, the unlabeled images are used to pretrain the backbone by a contrastive learning method, and the labeled images are used to fine-tune the network. The unlabeled images are then given pseudo-labels by the fine-tuned network and used for further training. In order to alleviate the uncertainty of the given pseudo-labels, we propose a Weight Cross-Entropy Loss (WCE-Loss) to suppress the influence of the samples with unreliable pseudo-labels in the training process. Experiment results on three prominent benchmark datasets for GER show the effectiveness of the proposed framework and its superiority compared with other competitive state-of-the-art methods.

**Keywords:** group emotion recognition; semi-supervised learning; contrastive learning; pseudo-labels





## 1. Introduction

Group emotion recognition (GER) usually classifies the overall emotion of a group image (or video) into three categories: positive, neutral, and negative. In addition to the problems such as occlusion and low resolution of human faces in group images, the performance of GER is also influenced by the interactions that exist between the individuals and groups and also between the environment and the group. These reasons make GER a more challenging task compared with individual emotion recognition. Research work in social psychology shows that group emotion contains "bottom-up" and "top-down" components, where the bottom-up component refers to the combination of individual emotions, such as people's expressions and actions, and the top-down component refers to the influence on individuals from the group or scene level [1]. How to extract and fuse the features of these components and improve the recognition accuracy is the major issue of GER research.

The rapid advancement of deep learning leads to many methods based on convolutional neural networks (CNNs) that were proposed to improve the performance of GER. In recent years, GER has started to be used in many important application scenarios, including image retrieval [2], detection of depression in people [3], image memorability prediction [4,5], public safety [6], human–computer interaction [7], etc.

However, the performance of current GER methods is still suffered from the limited number of labeled samples. Although there is a huge number of group images on the

Internet, manually annotating them with emotional labels is a labor-intensive and cost-expensive process. Typically, the annotation process usually requires every group image to be evaluated by three to five annotators, and the label also needs another round of proofreading before it is issued. This challenge limits the number of labeled samples for GER research.

Semi-supervised learning has been proven a promising method to leverage a large number of unlabeled data to improve the performance of a learning-based network [8–11]. However, the quality or reliability of learned features may suffer from the efficiency of the semi-supervised learning policy. Designing an efficient semi-supervised learning policy and improving the reliability of learned features for each specific application is still a challenging task.

In this work, we introduced semi-supervised learning for GER for the first time and proposed a contrastive learning-based group emotion recognition framework (SSGER). We trained the proposed network by contrastive learning to force it to extract similar semantic information from the scene and face regions of unlabeled images and assign a pseudo-label to the unlabeled image according to the knowledge that SSGER learns from labeled images. Image samples with either a real or pseudo label are used to update the parameters of the classification network. In order to compensate for the uncertainty of the pseudo-labels, we designed a Weight Cross-Entropy Loss (WCE-Loss) to balance the contributions of the image samples with real and pseudo-labels.

Our SSGER framework is composed of two networks, the SFNet and the FusionNet. The SFNet extracts the preliminary emotion information from the face and scene images, and the FusionNet generates a more comprehensive emotional feature of the group by fusing the emotion cues carried by the scene and faces. In order to make use of a large number of unlabeled group images, we proposed a training strategy composed of four stages. In Stage 1, we propose a contrastive learning method to pretrain SFNet in order to learn useful semantic information from the unlabeled data. In Stage 2, we use limited labeled data to train the SFNet and FusionNet. In Stage 3, we use the trained SFNet and FusionNet to assign unlabeled samples pseudo-labels. In Stage 4, the SFNet and FusionNet are further trained with both the samples with real labels and the samples with pseudo-labels. We propose a WCE-Loss to alleviate the influence of the samples with pseudo-labels in the backpropagation in Stage 4.

The remainder of this article is structured as follows: In Section 2, we give a brief review of the related works in GER and semi-supervised learning. In Section 3, we introduce our proposed method in detail. In Section 4, the experimental results and discussion are reported. Finally, we summarize our work in Section 5.

## 2. Related Work

### 2.1. Group Emotion Recognition

Many GER works focus on powerful facial features and scene features as they compose the most influential factors of group emotion. Some GER works also introduce the influence of other factors, such as objects [12] and the human skeleton [13]. Dhall et al. [14] proposed a GER framework to extract facial features from facial action units and used GIST and CENTRIST descriptors to characterize the emotion cues from the scene. Tan et al. [15] built three CNN models to learn emotional features from the aligned face, the unaligned face, and the whole image, respectively, and used an average fusion strategy to combine the output of these three models for the reason that the group emotion was regarded as a superposition of individual emotions. Surace et al. [16] proposed a GER method composed of neural networks and Bayesian classifiers, where the neural networks were used to analyze individual emotions based on a bottom-up approach, and the Bayesian classifiers were used to estimate the scene expression based on a top-down approach. Among the many fusion methods proposed to improve the model performance, the attention mechanism is one of the most popular techniques. Fujii et al. [17] used a visual attention mechanism to focus on the facial features of the major subjects in the group and suppressed

the facial features of others. Khan et al. [12] also proposed a regional attention mechanism to focus on more important persons. Some new methods were also introduced to improve the efficiency of feature fusion in GER. For example, long short-term memory (LSTM) was used to aggregate the features of scenes and faces [18–20]. Graph Neural Networks were also employed to fuse different emotional cues and exploit the underlying relations and interactions between the emotional cues [21]. Although the GER methods equipped with these fusion techniques have achieved promising experimental results, their performance still suffers from the limited number of labeled samples. In this article, we aimed to explore additional emotional features from the unlabeled image and improve the performance of group emotion recognition.

### 2.2. Contrastive Learning

Contrastive learning is a promising method for pretraining deep models. It helps the backbone network to learn efficient representations from unlabeled samples and serve the downstream tasks. For instance, Chen et al. [22] proposed a contrastive learning framework to capture similar features of the paired input images and facilitated the pre-trained network for image classification tasks. Similarly, He et al. [23] proposed a momentum contrast method to minimize the distance between the features learned from different augmented views of the same image and maximize the distance between the features learned from the same augmented views of different images. Benefiting from the Stop-Gradient operation, a simple contrastive method named SimSiam [24] has also been proposed, which can significantly reduce the batch size and the number of epochs compared with the existing methods. The SimSiam method also shows the effect of learning visual representation without semantic information. Contrastive learning was also used in face recognition [25] and face generation [26] tasks and achieved impressive performance. The above applications of contrastive methods show that contrastive learning has great potential to help the network capture efficient representations in pretraining. In this work, we also employed contrastive learning to pretrain the proposed network and help the proposed network extract effective semantic representations from group images and serve the group emotion recognition.

### 2.3. Semi-Supervised Learning

As a typical learning-based pattern recognition application, GER is expected to efficiently learn emotional features from training samples. The performance of current GER methods depends heavily on the number of labeled samples. Considering labeling the group images is labor-intensive and cost-expensive work, the sizes of the most widely used datasets for GER are still limited. We show the statistics of three widely used GER datasets, i.e., the Group Affective 2.0 (GAF2) dataset [27], the Group Affective 3.0 (GAF3) dataset [28], and the GroupEmoW dataset [29] in Table 1. The largest dataset shown in Table 1, the GroupEmoW, contains less than 16,000 images, which is much less than the number of samples in the datasets used by other recognition research such as ILSVRC [30], ImageNet [31], etc.

The technique of semi-supervised learning is a potential way to leverage the information of unlabeled samples and improve the performance of learning-based recognition models. Although semi-supervised learning methods were rarely used in GER, they have achieved promising experimental results in many learning-based recognition applications.

Semi-supervised learning uses a large number of unlabeled data to improve the performance of a learning-based network with the help of a limited number of labeled data. In semi-supervised learning, marking the unlabeled data with the pseudo-labels for training is one of the most popular strategies. These pseudo-label-based methods first use the limited labeled data to train an annotator that gives pseudo-labels to the unlabeled data [32]. Then, the data with real labels and pseudo-labels are used together to update the parameters of the proposed network. Pseudo-label-based methods have been widely used in the learning process to improve the performance of recognition networks. Xie et al. [8] proposed an iterative method to generate the pseudo-labels for the unlabeled data and used them to improve both the

accuracy and robustness of ImageNet models. Sohn et al. [9] proposed an integrated classifier to give unlabeled images with pseudo-labels and used them to improve the performance of a model on image classification tasks. Hao et al. [33] inferred pseudo-labels for unlabeled data with graph-based label propagation and promoted the ability of image change detection. Besides classification, semi-supervised learning is also used in many other applications, e.g., object detection [10,11], motion analysis [34], and multi-view models [35]. By marking pseudo-labels to the sample without labels, all the methods mentioned above [8–11,32–35] leverage the unlabeled data to help the updating of the learning-based network and improve the performance of recognition. However, the reliability or uncertainty of the marked pseudo-labels may influence the efficiency of the learning-based network. How to compensate for the uncertainty of the pseudo-labels is still an open question to semi-supervised learning methods.

**Table 1.** Statistics of the three datasets."-"means the test set is not available.

|  |  | **Positive** | **Neutral** | **Negative** | **Total** |
|---|---|---|---|---|---|
|  | train | 1272 | 1199 | 1159 | 3630 |
| GAF2 | val | 773 | 728 | 564 | 2065 |
|  | test | - | - | - | - |
|  | train | 3977 | 3080 | 2758 | 9815 |
| GAF3 | val | 1747 | 1368 | 1231 | 4346 |
|  | test | - | - | - | - |
|  | train | 4645 | 3463 | 3019 | 11,127 |
| Group-EmoW | val | 1327 | 990 | 861 | 3178 |
|  | test | 664 | 494 | 431 | 1589 |

In this work, we adapted the idea of semi-supervised learning and the technique of pseudo-labeling into a framework for group emotion recognition. We explored the emotional cues from the labeled and unlabeled group images and designed a new loss function to compensate for the uncertainty of pseudo-labels. Experimental results show that our framework is effective and has superior performance compared with other state-of-the-art methods.

## 3. Proposed Methods

The overview of the proposed SSGER is shown in Figure 1. The framework of SSGER consists of two networks, the SFNet and the FusionNet. We fed the SFNet with two inputs, i.e., the face images cropped from the group image and the scene image obtained from the group image. The SFNet extracts the preliminary emotion information from the face and scene images. We used FusionNet to fuse the emotional features extracted from the face and scene images and generate a more comprehensive emotional feature of the group. The group emotion is recognized based on the comprehensive features. In order to exploit the potential emotional cues from a large number of unlabeled samples of group images, we propose a unique training strategy of 4 stages. In Stage 1, we pretrained the SFNet with a contrastive learning method to extract semantic emotional information from the scene and face images of unlabeled data. In Stage 2, we used the limited labeled images to train the SFNet and FusionNet. In Stage 3, we froze the parameters of the SFNet and FusionNet and used them to give pseudo-labels to unlabeled images. In Stage 4, the SFNet and FusionNet were further trained with both the images with pseudo-labels and the images with real labels. In order to suppress the influence from the samples with unreliable pseudo-labels, we propose the Weight Cross-Entropy Loss, $L_{WCE}$, for the process of backpropagation in Stage 4.

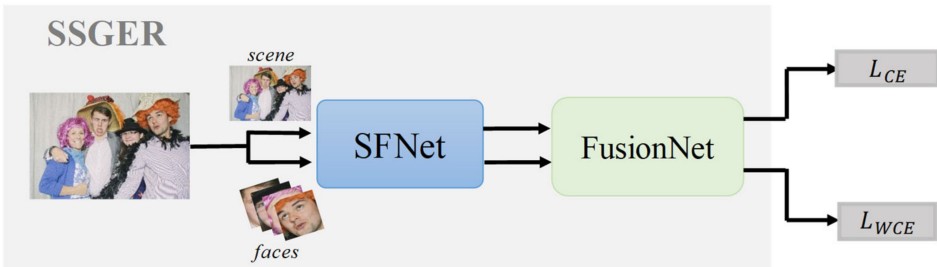

**Figure 1.** The overview of SSGER framework.

### 3.1. The SFNet

We used the ResNet-50 network as the backbone of the SFNet to capture the high-level CNNs features from the face image and scene image as the semantic emotional features of the group [36,37]. We segmented all the face regions from the group image and named them the face images, randomly cropped a region from the group image, and denoted it as the scene image. Each face image and scene image form an image pair. Then we fed the image pair into SFNet. The operations of feature extracting can be expressed by Equations (1) and (2),

$$x_i^s = \varphi(I_i^s) \tag{1}$$

$$x_{ij}^f = \varphi\left(I_{ij}^f\right) \tag{2}$$

where $\varphi(\cdot)$ denotes the process of the SFNet, $I_i^s$ is the scene image of the *i*-th group image, and $I_{ij}^f$ is the *j*-th cropped face image of the *i*-th group image. $x_i^s$ is the scene feature of the *i*-th image, and $x_{ij}^f$ is the *j*-th face feature of *i*-th group image.

### 3.2. The FusionNet

We designed the FusionNet to fuse the emotional features extracted from the face image and scene image. As shown in Figure 2a, the FusionNet composes an attention mechanism module and a prediction fusion module. The FusionNet takes the scene and face features as input.

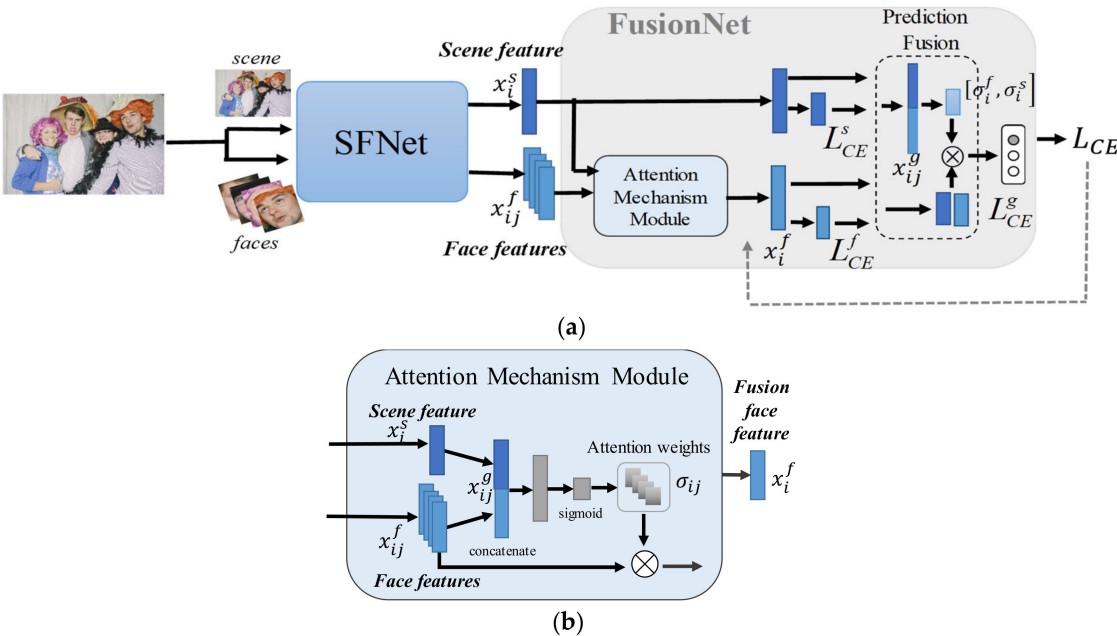

**Figure 2.** The structure and training flow of FusionNet. (**a**) The training flow of SFNet and FusionNet; (**b**) attention mechanism module in FusionNet.

In FusionNet, we employed an attention mechanism to focus on important facial features. In the attention mechanism module, as shown in Figure 2b, we concatenated the scene feature with each face feature separately, fed the concatenated feature into a fully connected layer, and used the Sigmoid function to learn the attention weight. The process of learning attention weights can be formulated by Equation (3),

$$\sigma_{ij} = \text{Sigmoid}\left(W_f \cdot x_{ij}^c + b_f\right) \tag{3}$$

where $\sigma_i$ denotes the attention weight of the *j*-th face feature in the *i*-th group image. $W_f$ and $b_f$ are weights and biases in the fully connected layer, respectively. $x_{ij}^c$ denotes the concatenated feature of the *i*-th scene feature and the *j*-th face feature. Sigmoid($\cdot$) is an operation of the sigmoid activation function. The group's emotional description by aggregating face features can be expressed as Equation (4),

$$x_i^f = \frac{\sum_{j=1}^N \sigma_{ij} \cdot x_{ij}^f}{\sum_{j=1}^N \sigma_{ij}} \tag{4}$$

where $x_i^f$ denotes the aggregated face feature for the *i*-th group image. Further, we fee = d the aggregated face feature and the scene feature into the fully connected layers and obtained the predictions of group emotion, $y_i^f$ and $y_i^s$, from the viewpoints of facial emotional information and scene emotional information, respectively.

With the two group emotion predictions from the viewpoints of the face and scene information, we present a module named Prediction Fusion to adaptively fuse the predictions of group emotion in the FusionNet. The process of prediction fusion can be formulated by Equation (5),

$$\hat{y}_i = \sigma_i^f y_i^f + \sigma_i^s y_i^s \tag{5}$$

where $\sigma_i^f$ and $\sigma_i^s$ are the fusion weights for the predictions exploited from the face and scene features, respectively. The fusion weights are required to satisfy the constraints that $\sigma_i^f$, $\sigma_i^s \geq 0$ and $\sigma_i^f + \sigma_i^s = 1$. In particular, the fusion weights are generated in a learning-based way. We concatenate the scene feature $x_i^s$ and the aggregated face feature $x_i^f$ into a comprehensive feature $x_i^g$, and fed the feature $x_i^g$ into a fully connected layer, followed by the Softmax function to learn the fusion weights for group emotion recognition. The learning process of fusion weights can be formulated by Equation (6),

$$[\sigma_i^f, \sigma_i^s] = \text{Softmax}\left(W_g \cdot x_i^g + b_g\right) \tag{6}$$

where Softmax($\cdot$) is the operation of the softmax activation function, and $W_g$ and $b_g$ are the weights and biases in the fully connected layer, respectively.

### 3.3. Training Process

In order to explore the potential emotional cues from unlabeled and labeled samples, we propose a training strategy for SSGER based on contrastive learning, pseudo-labeling, and pseudo-label suppressing. The training process is composed of four stages.

### 3.3.1. Stage 1: Pretraining SFNet with Contrastive Learning

How to leverage a large number of unlabeled images to improve the ability of the recognition network remains a challenge in GER. Based on the idea of contrastive learning [22–24], we used the SFNet to extract the semantic emotional information through the scene and face images cropped from the unlabeled image. The pretraining flow of the SFNet is shown in Figure 3.

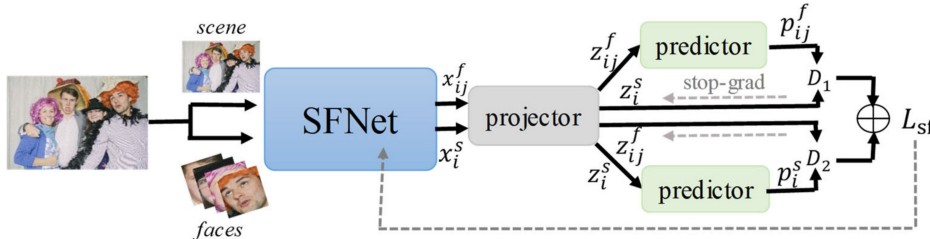

**Figure 3.** Pretraining flow of the SFNet.

In order to pretrain the SFNet, we constructed a three-layer MLP named projector to map $x_i^s$ and $x_{ij}^f$, which are the scene feature and face feature extracted by SFNet, by Equations (7) and (8),

$$z_{ij}^f = P\left(x_{ij}^f\right) \tag{7}$$

$$z_i^s = P(x_i^s) \tag{8}$$

where $P(\cdot)$ denotes the process of the projector, $z_i^s$ and $z_{ij}^f$ are the corresponding scene and faces feature after the mapping. Then, we use two predictors to map $z_i^s$ into the feature space of $z_{ij}^f$, and map $z_{ij}^f$ to the feature space of $z_i^s$. These two predictors are all made of a two-layer multilayer perceptron (MLP). The output of the predictors can be described as Equations (9) and (10),

$$p_{ij}^f = h\left(z_{ij}^f\right) \tag{9}$$

$$p_i^s = h(z_i^s) \tag{10}$$

where $h(\cdot)$ is the prediction function. Then, we calculated the negative cosine similarities between $p_{ij}^f$ and $z_i^s$, and between $p_i^s$ and $z_{ij}^f$, as expressed by Equations (11) and (12),

$$D_1\left(p_{ij}^f, z_i^s\right) = -\frac{p_{ij}^f}{\|p_{ij}^f\|_2} \cdot \frac{z_i^s}{\|z_i^s\|_2} \tag{11}$$

$$D_2\left(p_i^s, z_{ij}^f\right) = -\frac{p_i^s}{\|p_i^s\|_2} \cdot \frac{z_{ij}^f}{\|z_{ij}^f\|_2} \tag{12}$$

where $\|\cdot\|_2$ represents the $l_2$-norm. In order to prevent collapsing, we use the Stop-Gradient [24] method and cut off the branch without a predictor during the backpropagation process. The final loss function can be formulated by Equation (13),

$$L_{sf} = \frac{1}{2}D_1\left(p_{ij}^f, s(z_i^s)\right) + \frac{1}{2}D_2\left(p_i^s, s(z_{ij}^f)\right) \tag{13}$$

where $s(\cdot)$ denotes the process of the Stop-Gradient method. Therefore, we forced the SFNet to extract features with similar sematic representation from the scene and face images of the input image, no matter whether it is labeled or unlabeled.

3.3.2. Stage 2: Pretraining SFNet and FusionNet with Labeled Data

In Stage 2, all the labeled data were used to train the SFNet and FusionNet. We used a cross-entropy loss function, $L_{CE}^s$, to measure the recognition ability of the scene feature (see Figure 2a). $L_{CE}^s$ is formulated by

$$L_{CE}^s = -\log \frac{e^{W_{l_i}^T x_i^s}}{\sum_{k=1}^{C} e^{W_k^T x_i^s}} \tag{14}$$

where $l_i$ denotes the emotion label of the *i*-th image, $W_k$ denotes the parameters of the *k*-th classifier (fully connected layer) for the scene feature, and *C* is the number of categories of group emotion.

We also used the cross-entropy loss of face $L_{CE}^f$ to evaluate the recognition ability of the fused face features, which is formulated by Equation (15),

$$L_{CE}^f = -\log \frac{e^{V_{l_i}^T x_i^f}}{\sum_{k=1}^{C} e^{V_k^T x_i^f}} \tag{15}$$

where $V_k$ denotes the parameters of the *k*-th classifier for the fusion face feature.

We fused the predictions with the scene feature $x_i^s$ and the aggregated face feature $x_i^f$ to generate a unique prediction for group emotion. We designed a cross-entropy loss function, $L_{CE}^g$, to evaluate the recognition ability of the fused comprehensive emotional prediction. $L_{CE}^g$ is formulated by Equation (16),

$$L_{CE}^g = -\log \frac{e^{(\sigma_i^s W_{l_i}^T x_i^s + \sigma_i^f V_{l_i}^T x_i^f)}}{\sum_{k=1}^{C} e^{(\sigma_i^s W_k^T x_i^s + \sigma_i^f V_k^T x_i^f)}} \tag{16}$$

where $\sigma_i^s$ and $\sigma_i^f$ are the fusion weights for the predictions exploited from the face and scene features, respectively.

As shown in Equation (17), the overall loss function is the summation of the loss functions defined above, which evaluates the influence of the scene, faces, and the final fused features.

$$L_{CE} = L_{CE}^s + L_{CE}^f + L_{CE}^g \tag{17}$$

$L_{CE}$ is used in the backpropagation to update the parameters of the SFNet and FusionNet.

### 3.3.3. Stage 3: Giving Unlabeled Data with Pseudo-Labels

In order to exploit the emotional cues from the unlabeled data and improve the ability of feature representation of our proposed networks, in Stage 3, we annotated pseudo-labels to the unlabeled data with the networks trained in Stage 2.

We froze the parameters of the SFNet and FusionNet in this stage. For each unlabeled sample, we used the probabilistic output of the network as the soft pseudo-labels $y_{Pse}$ and recorded the index of the maximum probability of the output vector as the hard pseudo-labels $l_{Pse}$. With the soft and hard pseudo-labels, unlabeled samples can be involved in the backpropagation process of the further training of SSGER and facilitate the updating of network parameters.

### 3.3.4. Stage 4: Further Training of SSGER

In Stage 4, we performed a further training process on SSGER with the data with real labels and the data with pseudo-labels. Although we annotated the unlabeled data with pseudo-labels, the data with pseudo-labels may introduce an uncertain impact on the network training. We proposed the WCE-Loss in Stage 4 to suppress the impact of the unreliability of the pseudo-labels. The further training of SSGER is shown in Figure 4.

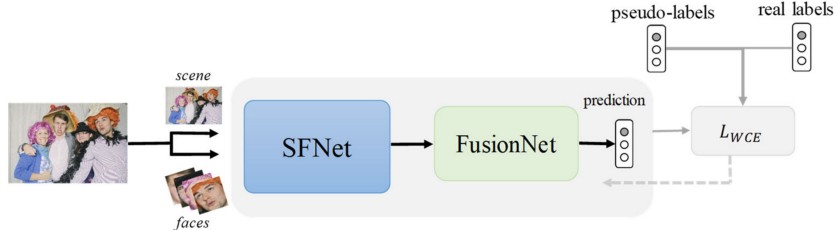

**Figure 4.** Further Training of SSGER.

In our work, we used the cosine similarity between the soft pseudo-label and the prediction output of the SSGER to measure the reliability of the pseudo-label. For any sampled annotated with a soft and a hard pseudo-label, a low value of the cosine similarity indicates a low reliability of the pseudo-label, and a small weight value is given to the sample corresponding to this pseudo-label in the backpropagation. On the contrary, a pseudo-label with a high value of cosine similarity is considered reliable, and a large weight value is given to the sample.

$$L_{WCE}^s = -\log \frac{e^{\alpha_i W_{l_i}^T x_i^s}}{\sum_{k=1}^{C} e^{\alpha_i W_k^T x_i^s}} \tag{18}$$

$$\alpha_i = \begin{cases} 1 & if\ labeled \\ \frac{y_{Pse}^T y_{Pre}}{\|y_{Pse}\|\|y_{Pre}\|} & if\ unlabeled \end{cases} \tag{19}$$

Equation (18) shows the WCE-Loss function when the scene features are evaluated, where $l_i$ denotes the hard label of group emotion in the $i$-th image. $W_k$ denotes the parameters of the $k$-th classifier for the scene feature, and $C$ is the number of categories of group emotion. As shown in Equations (18) and (19), we used $\alpha_i$ to compensate for the contribution of every training sample. If we use labeled data for training, $\alpha_i$ is set to 1. If we use data with pseudo-label for training, the value of $\alpha_i$ is set to equal the cosine similarity between the annotated soft pseudo-label, $y_{Pse}$, and the probabilistic output of prediction net, $y_{Pre}$.

Similarly, we can define the WCE-Loss functions $L_{WCE}^f$ for the fused face features and $L_{WCE}^g$ for the fused comprehensive features. The overall WCE-Loss can be expressed by Equation (20),

$$L_{WCE} = L_{WCE}^s + L_{WCE}^f + L_{WCE}^g \tag{20}$$

By this means, we are able to train the SSGER with both labeled and unlabeled data.

## 4. Experiments and Discussion

In order to evaluate the performance of the proposed SSGER method, we conducted extensive experiments on three popular GER datasets, i.e., GAF2, GAF3, and GroupEmoW. In these GER datasets, group emotion is annotated into three categories, i.e., positive, neutral, and negative. As the test sets of GAF2 and GAF3 are not accessible to the public, we used the training set to train the network and the validation set for test in the experiments. On the GroupEmoW dataset, we used the training set to train and the test set for testing. In order to evaluate the effectiveness of the proposed method in a semi-supervised scenario, we randomly selected the labeled samples with a given label rate (the ratio of the number of labeled samples to the total number of the training samples), and the rest of the samples in the training set were regarded as unlabeled data. We used the labeled and unlabeled samples for semi-supervised learning. As our work exploits emotional cues from the sources of face and scene for GER, for a fair comparison, we compared the proposed SSGER with six state-of-the-art baseline methods [12,16,17,38–40], which recognize group emotion using the information of face and scene. In this work, we used the metric of classification accuracy to measure the effectiveness of the GER methods.

### 4.1. Implementation Details

In Stage 1, faces in an image were cropped by multi-task cascaded convolutional neural networks (MTCNN) [41]. We set the maximum number of faces extracted from an image to 16 and resized the cropped faces to 224 × 224 pixels. Then, we performed operations of data augmentation, including random horizontal flip and Gaussian blur, to the resized faces. The whole image, which was also viewed as the scene input, was cropped and resized to 224 × 224 pixels. We also performed the same data augmentation on the whole image as we performed on the face image. We used a stochastic gradient descent

(SGD) optimizer with a learning rate of 0.05 in the pretraining process of SFNet and set the batch size to 256. In Stage 2, the data augmentation was the same as in Stage 1, except that there was no Gaussian blur. The Adam optimizer was used in this stage with a learning rate of $10^{-5}$ and the batch size of 4. In Stage 3, we froze the parameters of the SFNet and FusionNet and used the network to generate pseudo-labels for the unlabeled data. In Stage 4, the prediction net was trained by the Adam optimizer with a learning rate of $10^{-5}$ and the batch size of 4. For all stages, we trained the network for 100 epochs. All experiments were conducted on a Linux server with Intel Xeon CPU E5-2673 v4 2.30 GHz and GeForce GTX 2080Ti.

### 4.2. Comparison on Classification Performance

We compared our proposed method with other state-of-the-art methods on the GAF2, GAF3, and GroupEmoW datasets. In order to evaluate the efficiency of the proposed SSGER equipped with semi-supervised learning, we set the label rates to 5%, 10%, 30%, and 100% and performed comparison experiments. The case for the label rate of 100% is actually supervised learning. As there is not yet any reported GER method using semi-supervised learning, we used the performance of a ResNet-50-based network as the baseline for comparison.

Tables 2–4 show the proposed SSGER obtained superior overall accuracy compared to the baseline network (ResNet-50) on all three datasets. This advantage comes from the following reasons. Firstly, the proposed SSGER employs a contrastive learning-based network (SFNet) to extract semantic features from the faces and scene images of the unlabeled data. Secondly, the FusionNet trained with the labeled data in Stage 2 fuses the emotional features extracted from the face and scene images and generates an efficient description of group emotion. Thirdly, the SSGER uses the WCE-Loss to compensate for the uncertainty of the training data with pseudo-labels.

**Table 2.** Comparison of classification accuracy (in %) on the GAF2 dataset. The values in bold refer to the best result.

| Label Rate of Training Set | Semi-Supervised GER on the GAF2 Dataset | | | | |
| --- | --- | --- | --- | --- | --- |
| | Method | Positive | Neutral | Negative | Overall |
| 5% | ResNet-50 | 56.79 | 46.97 | 83.21 | 60.43 |
| | SSGER | 79.43 | 72.07 | 72.14 | 74.90 |
| 10% | ResNet-50 | 53.82 | 64.18 | 79.15 | 64.23 |
| | SSGER | 83.05 | 69.68 | 73.25 | 75.74 |
| 30% | ResNet-50 | 76.33 | 58.39 | 76.94 | 70.21 |
| | SSGER | 80.98 | 75.04 | 72.88 | 76.73 |
| 100% | ResNet-50 | 76.46 | 61.21 | **82.29** | 72.68 |
| | Surace + [16] | 68.61 | 59.63 | 76.05 | 67.75 |
| | Abbas + [38] | 79.76 | 66.20 | 69.97 | 71.98 |
| | Fujii + [39] | 75.68 | 69.64 | 77.33 | 74.22 |
| | Fujii + [17] | 78.01 | 72.92 | 76.48 | 75.81 |
| | SSGER | **85.38** | **84.49** | 60.89 | **78.51** |

**Table 3.** Comparison of classification accuracy (in %) on the GAF3 dataset. The values in bold refer to the best result, and the symbol of "-" indicates the result was not reported in the paper of the compared method.

| Label Rate of Training Set | Semi-Supervised GER on the GAF3 Dataset | | | | |
|---|---|---|---|---|---|
| | **Method** | **Positive** | **Neutral** | **Negative** | **Overall** |
| 5% | ResNet-50 | 80.54 | 64.99 | 43.62 | 65.19 |
| | SSGER | 83.40 | 63.38 | 66.69 | 72.37 |
| 10% | ResNet-50 | 72.70 | 72.81 | 52.72 | 67.07 |
| | SSGER | 82.60 | 71.64 | 64.66 | 74.07 |
| 30% | ResNet-50 | 85.46 | 62.14 | 58.81 | 70.57 |
| | SSGER | 83.23 | 66.81 | 72.38 | 74.99 |
| 100% | ResNet-50 | 86.89 | 61.62 | 67.75 | 73.52 |
| | Fujii + [39] | 72.12 | 69.51 | 71.52 | 71.05 |
| | Quach + [40] | - | - | - | 74.18 |
| | Fujii + [17] | 78.42 | 71.19 | 73.40 | 74.34 |
| | SSGER | **79.85** | **76.61** | **73.44** | **77.01** |

**Table 4.** Comparison of classification accuracy (in %) on the GroupEmoW dataset. The values in bold refer to the best result, and the symbol of "-" indicates the result was not reported in the paper of the compared method.

| Label Rate of Training Set | Semi-Supervised GER on the GroupEmoW Dataset | | | | |
|---|---|---|---|---|---|
| | **Method** | **Positive** | **Neutral** | **Negative** | **Overall** |
| 5% | ResNet-50 | 81.33 | 84.62 | 65.89 | 78.16 |
| | SSGER | 89.31 | 83.81 | 81.67 | 85.53 |
| 10% | ResNet-50 | 85.84 | 82.39 | 74.25 | 81.62 |
| | SSGER | 92.62 | 79.96 | 84.92 | 86.60 |
| 30% | ResNet-50 | 87.65 | 82.39 | 79.12 | 83.70 |
| | SSGER | 93.52 | 80.36 | 84.92 | 87.10 |
| 100% | ResNet-50 | 91.87 | 83.81 | 77.96 | 85.59 |
| | Khan + [12] | - | - | - | **89.36** |
| | SSGER | 94.13 | 85.22 | 84.22 | 88.67 |

As shown in Tables 2–4, our SSGER framework obtained superior or comparable overall accuracies on all datasets used, compared with other state-of-the-art methods. These results indicate that our method also performs well in the supervised scenario.

A more important result shown in Tables 2–4 is that the proposed SSGER used only 5% to 30% labeled samples to achieve superior or comparable overall accuracies obtained by the state-of-the-art methods with 100% labeled samples. For example, as shown in Table 2, the overall accuracies obtained by the SSGER were 74.90% when the label rate was 5%, 75.74% when the label rate was 10%, and 76.73% when the label rate was 30%, which were superior or comparable to the overall accuracies obtained by the state-of-the-art methods when all the labeled data were used. Tables 3 and 4 also demonstrate the advantage brought by the semi-supervised learning in the proposed SSGER.

### 4.3. Ablation Study

We argued that three techniques used in the SSGER make up its superior performance of group emotion recognition, including the contrastive learning for the pretraining of the SFNet, the process of marking unlabeled samples with pseudo-labels, and introducing WCE-Loss to compensate for the uncertainty of pseudo-labels. We performed ablation studies to investigate the effectiveness of these techniques in the proposed SSGER framework. The results of the ablation studies are shown in Table 5.

**Table 5.** Comparison on the performance of variant SSGER methods. The values in bold font denote the best accuracy. SSGER (w/o contrastive learning) denotes the framework without using contrastive learning, SSGER (w/o pseudo-label) denotes the framework without giving pseudo-labels, and SSGER (w/o WCE-Loss) denotes the framework without introducing WCE-Loss.

| Label Rate | Comparison on Overall Accuracies (in %) | | | |
|---|---|---|---|---|
| | Method | GAF2 | GAF3 | GroupEmoW |
| 5% | SSGER(w/o Contrastive Learning) | 69.91 | 68.74 | 82.63 |
| | SSGER(w/o Pseudo-Label) | 74.21 | 70.32 | 84.71 |
| | SSGER(w/o WCE-Loss) | 74.72 | 72.14 | 85.27 |
| | SSGER | **74.90** | **72.37** | **85.53** |
| 10% | SSGER(w/o Contrastive Learning) | 70.26 | 71.58 | 84.58 |
| | SSGER(w/o Pseudo-Label) | 73.72 | 72.60 | 85.53 |
| | SSGER(w/o WCE-Loss) | 75.59 | 73.72 | 86.09 |
| | SSGER | **75.74** | **74.07** | **86.60** |
| 30% | SSGER(w/o Contrastive Learning) | 74.11 | 72.02 | 85.90 |
| | SSGER(w/o Pseudo-Label) | 75.69 | 74.25 | 86.47 |
| | SSGER(w/o WCE-Loss) | 76.68 | 74.74 | 87.04 |
| | SSGER | **76.73** | **74.99** | **87.10** |
| 100% | SSGER(w/o Contrastive Learning) | 76.14 | 75.41 | 87.85 |
| | SSGER | **78.51** | **77.01** | **88.67** |

### 4.3.1. Ablation Study on Contrastive Learning

We compared the overall accuracies obtained with the SSGER with and without contrastive learning in Stage 1 on all three datasets. As shown in Table 5, the framework with the contrastive learning method obtained better overall accuracies than the framework without it. As the label rate decreases, the effectiveness of the framework using contrastive learning for pretraining becomes more significant.

### 4.3.2. Ablation Study on Pseudo-Labels

We also compared the SSGER with the framework without pseudo-labeling in the ablation study. Table 5 shows the SSGER using the pseudo-labeling method obtained better overall accuracies compared to the framework without it. In most cases in Table 5, the method with pseudo-labeling achieved a 1% to 2% improvement in accuracy, which benefited from using the unlabeled samples through the marked pseudo-labels. Experimental results also show that our method is effective in the case when few labeled samples were used. For example, when the label rate was 5% on the GAF2 dataset, the improvement was 0.69%.

### 4.3.3. Ablation Study on WCE-Loss

In this part, we performed experiments to demonstrate the effectiveness of the loss function of WCE-Loss on each dataset and at different label rates. Table 5 shows that in all cases, the framework with WCE-Loss obtained better overall accuracy than the framework without it. This performance benefited from the efficiency of the WCE-Loss as it compensates for the uncertainty introduced by the pseudo-labels and promotes both the labeled and pseudo-labeled samples to contribute the parameter updating.

### 4.4. Results of Different Label Rate

In order to investigate the influence of label rates on the classification performance, we performed experiments on the GAF2, GAF3, and GroupEmoW datasets at varied label rates and depicted the results in Figure 5.

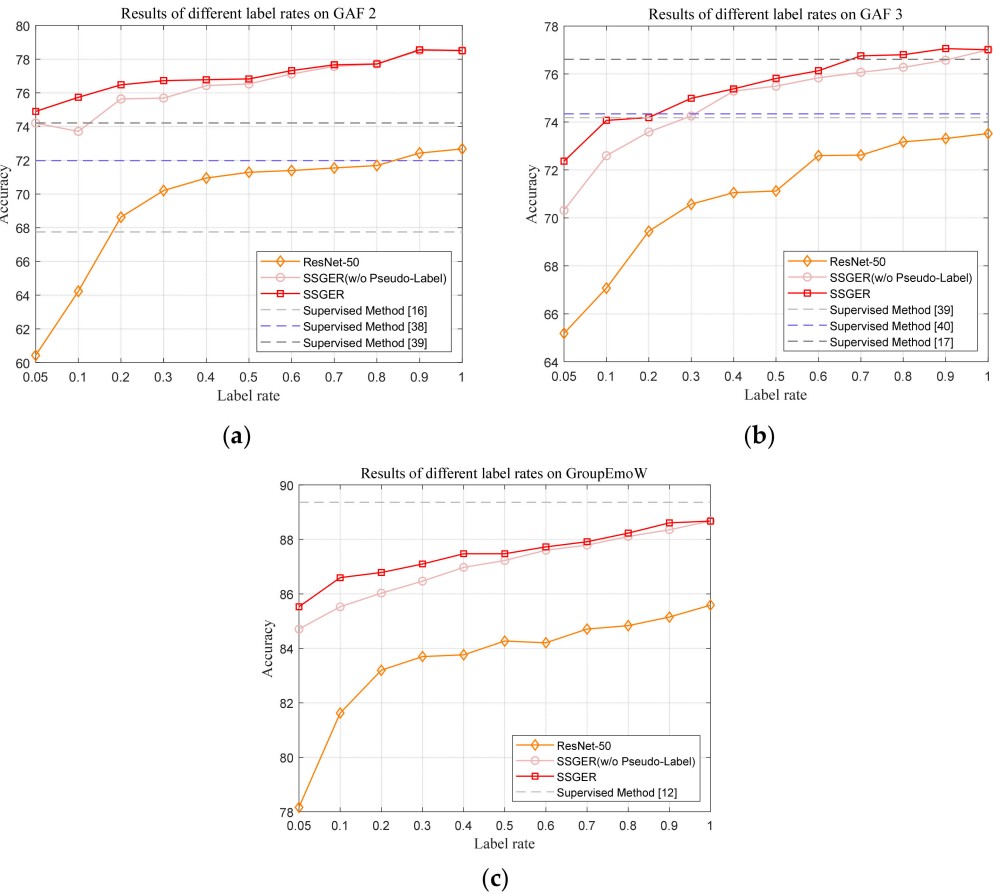

**Figure 5.** Accuracies of compared methods at different label rates on the GAF2, GAF3, and GroupE-moW datasets. The orange curve with diamonds shows the results of ResNet50 in the semi-supervised scenario. The pink curve with circles shows the results of our framework without using pseudo-labels. The red curve with rectangle shows the results of our SSGER framework using psedo-labels. The dashed lines denote the results of distinct state-of-art supervised methods in GER. (**a**) Results on GAF2; (**b**) results on GAF3; (**c**) results on GroupEmoW.

Figure 5a indicates that our SSGER framework achieved the best accuracy at distinct label rates on the GAF2 dataset. Even when the label rate was 0.05, our method still achieved better accuracy than the supervised method. Figure 5b shows that our semi-supervised framework obtained better overall accuracy than the ResNet50 and the framework without pseudo-labeling for all label rates. When the label rate was greater than 0.7, our SSGER framework obtained a better classification accuracy than the state-of-art supervised methods. The curves depicted in Figure 5b show that our framework also performed well with fewer labeled data on the GAF3 dataset. In Figure 5c, compared with the ResNet50 and the framework without pseudo-labeling, our framework also obtained a superior performance for each label rate on the three datasets.

Experiments results show that our SSGER framework is capable of extracting efficient, emotional features from both labeled and unlabeled samples and yields very competitive classification performance with limited labeled data.

## 5. Conclusions

In this work, we proposed a semi-supervised GER framework based on contrastive learning (SSGER) for datasets with limited labeled samples. We used unlabeled images to pretrain the SFNet with a contrastive learning method and used the labeled images to fine-tune the network. We used the fine-tune-trained network to give unlabeled images with pseudo-labels and designed the WCE-Loss to compensate for the uncertainty introduced by unreliable pseudo-labels. As the unlabeled images with pseudo-labels are used in

the training process, additional emotional features may be extracted from the unlabeled image and contribute to the recognition of group emotion. Experimental results on three datasets demonstrate the effectiveness of our SSGER framework. Comparison experimental results show that the proposed SSGER used only 5% to 30% labeled samples to achieve superior or comparable overall accuracies obtained by the state-of-the-art methods with 100% labeled samples.

**Author Contributions:** Conceptualization, J.Z., X.W. and D.Z.; methodology, J.Z. and X.W.; software, J.Z. and X.W.; validation, J.Z. and X.W.; formal analysis, X.W. and D.Z.; investigation, J.Z., X.W., D.Z. and D.-J.L.; resources, J.Z., X.W. and D.Z.; data curation, J.Z., X.W. and D.Z.; writing—original draft preparation, J.Z. and X.W.; writing—review and editing, D.Z. and D.-J.L.; visualization, J.Z. and X.W.; supervision, D.Z. and D.-J.L.; project administration, D.Z.; funding acquisition, D.Z. All authors have read and agreed to the published version of the manuscript.

**Funding:** This work was supported by the National Natural Science Foundation of China (62173353), Science and Technology Program of Guangzhou, China (202007030011).

**Data Availability Statement:** The Group Affective 2.0 (GAF2) and Group Affective 3.0 (GAF3) datasets are available in the Group Level Affect project, https://sites.google.com/view/grouplevelaffect/home (accessed on 9 November 2022). The GroupEmoW Dataset is accessible at GitHub, https://github.com/gxstudy/Graph-Neural-Networks-for-Image-Understanding-Based-on-Multiple-Cues (accessed on 9 November 2022). The data generated during the current study are available from the corresponding author on reasonable request.

**Conflicts of Interest:** The authors declare no conflict of interest.

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
