# Peer review of "Semi-Supervised Group Emotion Recognition Based on Contrastive Learning"

_electronics, doi:10.3390/electronics11233990_

Round 1

Reviewer 1 Report

Based on a careful analysis, I can formulate the following remarks:

1) The aim of this article, based on the authors’ scrupulous investigations, is developing a novel semi-supervised Group Emotion Recognition (GER) framework based on contrastive learning (SSGER) for datasets with limited labelled samples. They use unlabelled images to pre-train the SFNet with a contrastive learning method, and use the labelled images to fine-tune the network.

2) The topic represents in my opinion a relevant approach of the proposed theme in the field of the speech emotion recognition modelling, based on meticulous theoretical and investigations, correlated with experimental results.

3) In comparison with other published material, the authors' contribution adds to the subject area a new approach/methodology, with several significant contributions, such as:

·         They propose a semi-supervised group emotion recognition (GER) framework based on contrastive learning to learn efficient features from both labelled and unlabeled images.

·         In the proposed method, the unlabelled images are used to pre-train the backbone by a contrastive learning method and the labelled images are used to fine-tune the network;

·         Then, the unlabeled images are given pseudo-labels by the fine-tuned network and used for further training;

·         In order to alleviate the uncertainty of the given pseudo-labels, they propose a Weight Cross-Entropy Loss (WCE-Loss) to suppress the influence of the samples with unreliable pseudo-labels in the training process;

·         The performed experiments, by their results on three prominent benchmark datasets for GER show the effectiveness of the proposed framework and its superiority compared with other competitive state-of-the-art methods;

·         The analyzed, very significant (high) numbers of references, which can represent a solid database for further investigations, underline the authors’ involvement in the topic;

·         One has to underline the fact that the obtained results are very promising and they were better than the other reported ones from the literature.

4) It is well-known fact that Group Emotion Recognition (GER) usually classifies the overall emotion of a group image (or video) into three categories: positive, neutral and negative.

In addition to the problems such as occlusion and low resolution of human faces in-group images, the performance of GER is also influenced by the interactions existed between the individuals and groups, and between the environment and the group as well.

These reasons make GER a more challenging task compared with individual emotion recognition.

Research work in social psychology shows that group emotion contains "bottom-up" and "top-down" components, where the bottom-up component refers to the combination of individual emotions such as people's expressions and actions, and the top-down component refers to the influence on individuals from the group or scene level.

How to extract and fuse the features of these components and improve the recognition accuracy is the major issue of the GER research.

The rapid advancement of deep learning leads to many methods based on Convolutional Neural Networks (CNNs), proposed to improve the performance of GER.

In recent years, GER has started to be used in many important application scenarios, including image retrieval, detection of depression in people, image memoryability  prediction, public safety, human computer interaction and so on.

However, the performance of current GER methods still suffer from the limited number of labelled samples. Although there is a huge number of group-images on the Internet, annotating them with emotional labels manually is a labour-intensive and cost-expensive process. Typically, the annotation process usually requires every group image to be evaluated by 3 to 5 annotators and the label needs another round of proofread before it is issued.

This challenge limits the number of labelled samples for GER research.

Semi-supervised learning proved to be a promising method leveraging a large number of unlabelled data to improve the performance of a learning-based network.

However, the quality or reliability of learned features may suffer from the efficiency of the semi-supervised learning policy. Designing an efficient semi-supervised learning policy and improving the reliability of learned features for each specific application is still a challenging task.

In this contribution the authors propose a semi-supervised group emotion recognition framework (SSGER) based on contrastive learning.

They train the proposed network by contrastive learning to force it to extract similar semantic information from the scene, face regions of unlabeled images, and assign a pseudo-label to unlabeled image according to the knowledge that SSGER learns from labelled images. In order to updating the parameters of classification network, they used image samples with either a real or a pseudo label.

To compensate for the uncertainty of the pseudo-labels, they designed a Weight Cross-Entropy Loss (WCE-Loss) to balance the contributions of the image samples with real and pseudo labels.

Their SSGER framework was composed of two networks, the SFNet and the FusionNet; the SFNet extracts the preliminary emotion information from the face and scene images, and the FusionNet generates a more comprehensive emotional feature of group by fusing the emotion cues carried by the scene and faces.

To make use of the large quantity of unlabeled group images, the authors propose a training strategy composed of four stages:

·         a contrastive learning method to pre-train SFNet in order to learn useful semantic information from the unlabeled data;

·         they use limited labelled data to train the SFNet and FusionNet;

·         they use the trained SFNet and FusionNet to assign unlabeled samples pseudo-labels;

·         the SFNet and FusionNet are further trained with both the samples with real labels and the samples with pseudo-labels; the authors propose a Weight Cross-Entropy Loss to alleviate the influence of the samples with pseudo-labels in the back-propagation.

In order to evaluate the performance of the proposed SSGER method, they conducted extensive experiments on three popular GER datasets, i.e., the Group Affective 2.0 (GAF2) dataset, the Group Affective 3.0 (GAF3) dataset, and the GroupEmoW dataset. In these GER datasets, group emotion is annotated into three categories, i.e., positive, neutral, and negative. As the test sets of GAF2 and GAF3 are not accessible to the public, they used the training set to train the network and the validation set for test in the experiments. On the GroupEmoW dataset, they used the training set to train and test set for testing. To evaluate the effectiveness of the proposed method in semi-supervised scenario, they randomly selected the labelled samples with a given label rate (the ratio of the number of labelled samples to the total number of the training samples), and the rest samples in the training set were regard as unlabeled data. The authors used the labelled and unlabeled samples for the semi-supervised learning. As their work exploits emotional cues from the sources of face and scene for GER, for fair comparison, they compared the proposed SSGER with six state-of- the-arts baseline methods, which recognize group emotion using the information of face and scene.

In this contribution, the authors used the metric of classification accuracy to measure the effectiveness of the GER methods.

Experiments results show that their SSGER framework is capable of extracting efficient emotional features from both labelled and unlabelled samples, and yields very competitive classification performance with limited labelled data.

5) In my opinion, the presented conclusions are suitable related to their research results and prove that they reached the proposed goal.

6) The references in my opinion are very appropriate and their number underlines the scrupulosity of the authors.

7) In this paper, the graphical illustration is well conceived and realized and consequently they contribute to a better understanding of the performed theoretical investigations as well as to underlining the experimental validation of the proposed methodology.

I encourage publishing in a new contribution their further results.

Reviewer 2 Report

1)     Write (GER) in line 28

2)     Line 53: [8,9,10,11]. -> [8-11]. ; check and revise lines 103, 160, 253, 254

3)     Shift the full form of WCE from lines 75 to 63 when it appears for the first time

4)     Clearly state the novelty in the Section 1, whether this paper is the first time in the world or it is modification from previous paper? Summary from lines 157 – 170 regading the novelty and the difference to related works

5)     The related works in Section 2.1 only show that the performances are worse. Connect it to the novelty of the proposed network

6)     Line 159: models[35]. -> models [35].

7)     Fill the blank space in lines 171 – 177

8)     Line 183: face images and scene image. -> face and scene images.

9)     Line 197: write detailed definition of “semantic”… In the 6G or 7G, semantic communications are studied

10)  Use \cdot inside the brackets line 205

11)  Typo in line 223: opertation

12)  Typo in line 224: aggreagting

13)  Replace dot with comma in line 280, check others. It should avoid the use of dot before the equation, because the equations are part of the complete sentence. Revise lines 266, 270, and others

14)  Write the full form of MLP in line 258

15)  Replace stopgrad with a math symbol in (13)

16)  Line 285: be consistent with writing “Figure” or “Fig.”

17)  Line 343: be consistent to use the acronym of WCE

18)  The full form of acronyms in Lines 350 – 351 should be shifted to when they appear for the first time above

19)  Write the full form of MTCNN

20)  Tables 2, 3, and 4 -> Tables 2 4

21)  Replace the image in Figure 5 with higher quality with vector format

22)  Provide the quantity for “effectiveness” in the conclusion part
